# Parental behavior, adult attachment, and DNA methylation of the MT2 oxytocin receptor gene region – The moderating role of neuroticism

**Laura Geißert** [ORCID]*, **Juergen Hennig**

Deparment of Personality and Biological Psychology, Justus-Liebig-University Giessen, Giessen, Germany

* laura.geissert@psychol.uni-giessen.de

## Abstract

Parental behavior, especially in childhood, affects the child's development in numerous ways. Over the last decade, the aim to get a deeper understanding of how early experiences influence behavior later in life has led to an increased popularity of epigenetic studies. Several studies focused on negative childhood experiences, increased methylation at different oxytocin receptor gene sites, and deficits in social behavior in adolescence or adulthood. The current study focused on the role of parental behavior, personality, and methylation of the MT2 region in the oxytocin receptor gene on insecure attachment styles in young adulthood. A total number of $N = 71$ students (55 females, one non-binary) completed an online survey and provided cell material (buccal cell swaps) for methylation analysis. Parental behavior was measured with the Parental Bonding Instrument (PBI), personality with the NEO Five-Factor Inventory (NEO-FFI), and adult attachment with the Attachment Style Questionnaire (ASQ). Results showed a moderating effect of neuroticism on the relation between maternal care and methylation of the MT2 region: higher maternal care was associated with lower methylation levels but only among participants with low neuroticism scores. No association of methylation with anxious or avoidant attachment was observed and no effect of paternal care at all. The results emphasize the model of early environmental influences on behavior in respect to changing gene activity and will be discussed with respect to the MT2 region and early life experiences on the one, and the association with personality on the other hand.

## Introduction

With respect to social behavior, the neuropeptide oxytocin plays a significant role across species. It has been associated with attachment and social exploration but also with social deficits, for example, in autism spectrum disorders or depression (for an overview see Meyer-Lindenberg et al. [1]). For a long period of time, the focus regarding molecular genetics of oxytocin has been on

**Data availability statement:** The datasets presented in this study can be found in online repositories. The names of the repository and accession number can be found at the following link: https://doi.org/10.6084/m9.figshare.26830564.

**Funding:** The author(s) received no specific funding for this work.

**Competing interests:** The authors have declared that no competing interests exist.

single-nucleotide-polymorphisms in the oxytocin receptor gene (OXTR), most prominently the rs53576, located in the third intron, see Li et al. [2].

Over the last decade, the aim to get an even deeper understanding of how oxytocin affects social behavior has led to epigenetic studies, mostly on DNA methylation. Epigenetics refers to changes in gene transcription, which can result in either silenced or enhanced gene expression, without changes of the DNA sequence itself. Epigenetic changes are reversible and are hypothesized to rely amongst others on environmental influences and individual experiences [3, 4]. Methylation in particular changes the functional link between DNA and proteins, which can influence the transcription rate [5]. It takes place on cytosines (C) followed by guanines (G), known as Cytosine-phosphate-Guanine (CpG) sites. CpG rich regions in the genome are known as CpG islands and are often located in the promoter region of a gene [5]. CpG islands are > 200 base pairs (bp) in length and have a GC content of over 50% [6]. CpG islands tend to be unmethylated, methylation of these areas often leads to reduced DNA transcription, which can result in altered protein synthesis [4].

When it comes to epigenetic studies focusing on the OXTR, the MT2 region, a 406 bp long region covering parts of exon 1 and intron 1, is the most prominent [7]. Kusui et al. [8] were the first to detect the MT2 region in the CpG island of the OXTR. This region, when methylated, appears to be responsible for most of the suppression of the OXTR function. Therefore, MT2 is attributed to be functionally significant. The MT2 region contains 26 CpG sites. Since most studies focused on either all 26 CpG sites or only selected ones, which were often not specified, results are heterogenous [9]. However, if studies did focus on specific locations, CpG site −934 (relative to transcription start site +1, [10]) is amongst the most prominent. Danoff and colleagues [7] emphasized that selected CpG sites for epigenetic studies should be relevant to biological function. Therefore, they conducted a study on prairie voles as an animal model and showed that the CpG sites within the MT2 region correlate highly with each other [11]. Using exploratory graph analysis, they classified the CpG sites into three distinct groups depending on their covariance of methylation values. The categorization reflects the physical structure of the MT2 region: 5' sites, sites in the middle, and 3' sites. The 3' sites, which contain the sites −934, −924 and −901 (see Fig 1), seem to be most affected by experiences early in life [11].

Several studies have shown associations of MT2 hypermethylation, reduced OXTR activity, and different diseases, e.g., anorexia nervosa [12] or depression [13]. Furthermore, negative experiences in childhood and reduced maternal care have also been associated with decreased OXTR function [14, 15]. However, this effect is small, $r = .02$, as shown in a recent meta-analysis by Ellis et al. [9]. Research linking MT2 (hypo-) methylation and positive experiences is rare. An animal study in prairie voles by Perkeybile et al. [16] showed a direct link between increased early parental care and decreased DNA methylation in the MT2 region, which resulted in increased oxytocin receptor density. Krol, Moulder et al. [14] conducted a human study on children and their mothers. They showed that higher maternal care was associated with decreased methylation, which in turn up-regulated the oxytocin system, making infants more susceptible to social cues. Moreover, OXTR DNA methylation was

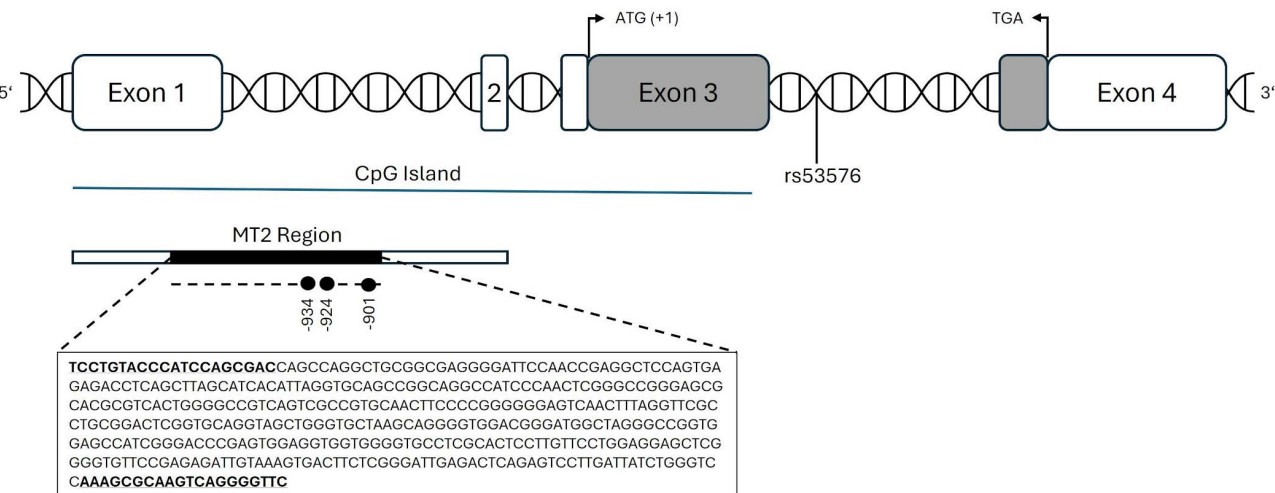

**Fig 1. Schematic overview of the MT2 region within the CpG island of the oxytocin receptor gene.** The protein-coding region is shown in gray (ATG = transcription start site; TGA = stop codon). The DNA sequence for the MT2 region is displayed with primer binding sites in bold. Functional relevant CpG sites are indicated by filled circles, numbering is relative to transcription start site (+1).

reported to be related to attachment behavior later in life. One study found that decreased methylation levels at CpG site −934 were associated with decreased levels of self-reported attachment anxiety and increased levels of attachment avoidance [17]. Ein-Dor et al. [18], however, found a positive association between OXTR methylation and attachment avoidance and no association with attachment anxiety. To the best of our knowledge, no further studies have examined the relationship between OXTR methylation and adult attachment.

It is well known that experiences early in life lay the foundation for attachment behavior in adolescence and adulthood [19]. According to the intergenerational transmission of attachment model, parents with secure attachment styles themselves operate as a secure base for their children, which increases the likelihood of their children being securely attached as well [20]. However, if this reflects environmental or genetic effects is an open question. Feeney [21] conducted a study, comparing parents and their children. Parents, who described themselves as more securely attached, had children with more secure attachment styles. This association was stronger for mothers compared to fathers, reflecting the more prominent role of mothers in parenting at the time of the study. Feeney highlighted parallels to the model of intergenerational transmission by IJzendoorn & Bakermans-Kraneburg. However, this conclusion has to be drawn with caution, because the intergenerational transmission of attachment goes beyond the attachment pattern of the parents. Other factors, for example psychosocial risk factors or the age of the children, may vary the strength of the association, for a revised model see Verhage et al [22]. A study by Tanaka et al. [23] found a positive association between early parental care and secure attachment in Japanese university students. In a cross-cultural study with university students from Singapore and Italy, high levels of overprotection in combination with low levels of care were associated with anxious as well as avoidant attachment in both cultures. Avoidant attachment was only associated with maternal care, whereas anxious attachment was associated with both, maternal and paternal care. Wilhelm et al. [24] compared the four attachment styles (secure, preoccupied, fearful, and dismissing) measured by the Relationship Questionnaire [25] with four perceptions of parenting behavior (optimal parenting, affectionate constraint, affectionless control, and neglectful parenting) measured by the Parental Bonding Instrument (PBI, [26]). They found that lower parental control corresponded with an increased likelihood of an adult attachment style with a positive view of self (secure for females and dismissing for males).

At this point it can be summarized that both epigenetic chances and parental behavior may have independent or interactive influences on adult attachment behavior. However, another aspect should be mentioned. Both influences may have different effects depending on individual differences at an individual level. Therefore, personality should also be addressed.

Personality and early experiences emerge as the two factors contributing to a gene x environment interaction. This concept states that people with different genetic make-ups can respond differently to environmental influences [27]. The Five Factor Model [28] is the most common theory of personality structure. The five factors are Extraversion, characterized by active, enthusiastic, and outgoing behavior; Agreeableness, generous, trusting, and appreciative behavior; Conscientiousness, efficient, reliable, und responsible; Neuroticism, displaying anxious, unstable, und tense behavior; and Openness to experiences, curious, insightful, and wide interested [28]. Neuroticism, which is approximately 40% heritable [29], is associated with increased environmental sensitivity [29, 30]. People high in neuroticism are more sensitive to the environment, due to enhanced perception and processing of information. This was shown for negative as well as positive influences [31].

Several studies looked at the interplay between perceived parental behavior and the child's personality. For example, Takahashi et al. [32] revealed that the perceived parenting style was associated with the child's neuroticism scores. Compared to optimal parenting (high care and low overprotection), people describing their parents' behavior as affectionless control had higher neuroticism scores. These results were replicated in a recent study, which illustrates that people who experienced childhood adversity exhibited higher neuroticism scores [33]. This leads to the assumption, that neuroticism seems to moderate the sensitivity with which people react to social experiences [34]. However, the path from perceived parenting towards neuroticism is not unidirectional but rather bi-directional: the child's personality can influence the way parents behave towards their children (gene x environment correlation) and vice versa [35].

In line with the model of gene x environment interactions, the present study, exploratory and preliminary in nature, examines the association between parental behavior, MT2 methylation, and attachment styles later in life. To narrow the gap regarding parental behavior, maternal as well as paternal care was considered. Due to the heterogeneous findings with respect to attachment avoidance and attachment anxiety, each component was used as the dependent variable. Therefore, the following model was hypothesized:

If perceived parental care in childhood is low, this may be accompanied by changes in OXTR MT2 methylation levels. As a consequence, avoidant and anxious attachment may increase. Therefore, it is assumed that the relationship between perceived parental care and attachment is associated with MT2 methylation levels. Moreover, neuroticism, a personality factor associated with salience of social stimuli, is expected to moderate both relationships, between perceived parental care and MT2 methylation levels, and between perceived parental care and adult attachment. Subjects high on neuroticism will more intensively perceive parental behavior related to care and, consequently, may be more susceptible for epigenetic effect. Moreover, the same pattern of perception will also be relevant for the association between perceived parental care and adult attachment.

The following central hypotheses were tested:

Higher levels of perceived care in childhood are associated with decreased levels of [a] attachment avoidance and [b] attachment anxiety, both being associated with methylation rates in the OXTR MT2 region. This is expected for [c] maternal and [d] paternal care separately. Additionally, it is assumed that neuroticism moderates both the effect between parental care and MT2 methylation levels, as well as between parental care and attachment anxiety (see Fig 2).

## Materials and methods

### Participants

Data were collected from two cohorts of students that started studying psychology in 2021 and 2022 at the Justus-Liebig-University in Giessen. Participants were recruited in the lecture Personality Psychology. Data collection was part of a large-scale study. Only relevant parts will be described here.

  

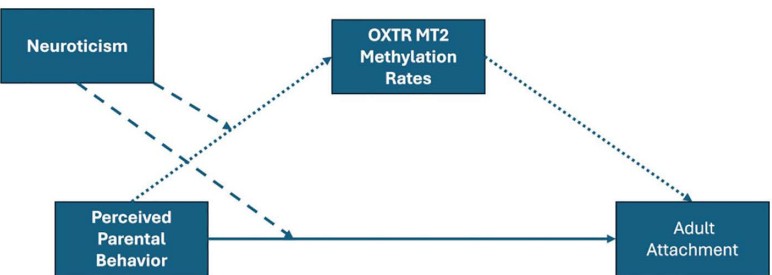

**Fig 2. Proposed conditional effects model.** The interplay between perceived parental behavior and adult attachment in consideration of neuroticism and oxytocin receptor gene MT2 methylation rates. *Note:* Moderation analysis indicated by dashed line. Mediation analysis indicated by dotted line.

An initial sample of $N = 367$ students started participating in the study, there were no exclusion criteria. Due to incomplete questionnaires ($n = 197$), no buccal swaps ($n = 23$), no sufficient sample DNA ($n = 68$), and non-detectable methylation rates ($n = 8$), the final sample consisted of $N = 71$ participants ($n = 55$ females, $n = 1$ non-binary). The mean age was $M = 21.15$, $SD = 3.91$ with a range between 18 and 48 years of age. About 60% of the participants were single ($n = 42$). All participants were Caucasians.

## Procedure

Three questionnaires relevant for the present study were embedded in a large-scale online survey. The online survey could be completed between December 08, 2021, and July 15, 2022, and from November 14, 2022, until May 31, 2023. Other questionnaires measured, for example, empathy, life events, or character strengths. In total, completion of the package took about two hours. It was possible to take breaks in between.

Buccal cell swaps were collected within the respective cohort at fixed dates, May 16, 2022, and November 07, 2022.

Participation was voluntary and was reimbursed with research participation credits. Written informed consent was received from every participant prior to the study. The study complied with the Declaration of Helsinki and was approved by the local ethics committee of the University of Giessen, Department of Psychology (application number: 2021−0017).

## Measures

### Questionnaires

Parental behavior was measured with the German version [36] of the PBI [26]. The PBI consists of two questionnaires, one regarding the mother and one regarding the father, with 25 identical items each. It investigates retrospective perception of Parental Care (12 items) and Overprotection (13 items) in the first 16 years of life. The German version of the PBI has been validated for the two-factor (care and overprotection) as well as the three-factor (care, denial of psychological autonomy, and encouragement of behavioral freedom) solution [36]. In the present study the two-factor solution was used. Only participants who completed the questionnaire for both, mother and father, were considered. All scales showed good internal consistency (Cronbach's α): Maternal Care α = .907, Maternal Overprotection α = .927, Paternal Care α = .939, and Paternal Overprotection α = .918.

Attachment was measured using the German version [37] of the Attachment Style Questionnaire (ASQ, [38]). The ASQ consists of 40 items in total, resulting in five scales: Confidence (eight items), Discomfort with Closeness (ten items), Need for Approval (seven items), Preoccupation with Relationships (eight items), and Relationships as Secondary (seven items). Discomfort with Closeness and Relationships as Secondary can be summarized as avoidant attachment. Anxious attachment, on the other hand, consists of Need for Approval and Preoccupation with Relationships. Again, for the given

sample, all ASQ scales showed good internal consistency (Cronbach's α): Confidence α = .911, Discomfort with Closeness α = .847, Need for Approval α = .748, Preoccupation with Relationships α = .749, and Relationships as Secondary α = .761.

To assess the big five personality factors, the German version [39] of the NEO Five-Factor-Inventory (NEO-FFI, [40]) was used. The NEO-FFI consists of a total of 60 items, 12 items per personality factor. The scales display good internal consistency (Cronbach's α), except for Agreeableness: Neuroticism α = .806, Extraversion α = .837, Openness for Experiences α = .826, Agreeableness α = .623, and Conscientiousness α = .881.

### DNA sampling

Participants provided buccal cell swaps for further analysis. According to Theda et al. [41] methylation rates in buccal cell swabs are ectodermal in origin, which means that the cell content is more similar to brain tissue than the cell content of blood samples. Furthermore, buccal swabs have less between-sample variation than saliva samples. Thus, buccal swabs are considered an appropriate source for methylation analysis [41], especially in the OXTR [42].

Purification of genomic DNA was performed with a standard commercial kit (QIAamp DNA Mini Kit, catalog no. 51306; QIAGEN, Hilden, Germany) in a QIACube (QIAGEN; Hilden, Germany), according to the manufacturer's protocol. Following, the DNA was quantified using NanoPhotometer™ Pearl P 300 (Implen GmbH; Munich, Germany). Only samples with an A260/A280 ratio between 1.7 and 1.9 were processed. The samples were stored at 4°C until bisulfite conversion.

### Bisulfite conversion

Bisulfite conversion was performed with EpiTect Fast DNA Bisulfite Kit (QIAGEN, catalog no. 59826; Hilden, Germany) in a QIACube (QIAGEN; Hilden, Germany). The low-concentration approach in the setup of the bisulfite reactions was used for all samples. For thermal cycling, the Mastercycler 5333 (eppendorf; Hamburg, Germany) was used. Bisulfite conversion thermal cycler conditions were slightly modified according to recommendations in the manual: the 60°C cycle time was extended to 20 minutes, resulting in a total thermal cycler time of 60 minutes. The second step, clean-up of bisulfite converted DNA, was performed in the QIACube according to the manufacturer's protocol. Following bisulfite treatment, methylated Cs at CpGs remain Cs, whereas unmethylated Cs at CpGs translate into uracil, later thymine (T).

### Primer design

Methyl Primer Express™ Software v1.0 (Applied Biosystems; Foster, USA) was used to manually design primers covering a 406 bp region of the OXTR, termed MT2. This region has proven to be functionally relevant [8]. The MT2 region is located on chromosome 3 (GRCh38: 8 769 033–8 769 438) and contains 26 CpG sites. The following primer pair for bisulfite converted DNA was used: 5'- GGAATTTTTGATTTGYGTTTT −3' (forward) and 5'- TCCTATACCCATCCAACRAC −3' (reverse) (Thermo Fisher Scientific, Waltham, Massachusetts). Both primers included M13 universal tails for amplification. Primers should contain a maximum number of one CpG site. The ensemble.org homepage (https://www.ensembl.org/Homo_sapiens/Location/Variant/Table?r=3:8769034-8769438) was used to check for possible single-nucleotide-polymorphisms within the complementary DNA region of primer binding.

### Amplification & cycle sequencing

For amplification the BigDye© Direct Cycle Sequencing Kit (Thermo Fisher Scientific, catalog no. 4458687; Waltham, Massachusetts) was used. Forward and reverse primers were treated separately. The manufacturer's protocol was modified as follows: 3 ng/µl instead of 4 ng/µl of bisulfite converted DNA was used. Furthermore, 0.5 µl M13-tailed PCR primer (10µM) mix per primer was added; therefore, the amount of distilled water was increased to 3.5 µl per reaction mix [43]. Thermal cycler (Mastercycler 5333, eppendorf; Hamburg, Germany) time was modified with respect to selected primers. Temperature and time are displayed in Table 1.

**Table 1. Thermal cycler time and temperature for amplification.**

| Temperature | Time | |
|---|---|---|
| 95° | 5 min | |
| 95° | 10 sec | 5 cycles |
| 62° | 2 min | |
| 72° | 3 min | |
| 95° | 10 sec | 35 cycles |
| 65° | 30 sec | |
| 72° | 1 min | |
| 4° | ∞ | |

Cycle sequencing was performed with the indicated amount of DNA and BigDye® Direct Primer (forward and reverse separately). Thermal cycling (Mastercycler 5333, eppendorf; Hamburg, Germany) was adapted as displayed in Table 2.

## Purification & sequencing

Purification was performed with the DyeEx® 2.0 Spin Kit (QIAGEN, catalog no. 63206; Hilden, Germany), since mechanical purification was proven to be superior to chemical purification for the given samples. After pipetting samples on to a plate, the plate was spun in a swinging-bucket centrifuge (Centrifuge 5910 Ri; eppendorf, Hamburg, Germany).

Sanger Sequencing by capillary electrophoresis was carried out on a SeqStudio (Life Technologies Holding; Singapore) according to manufacturer's protocol. For specification, Medium_Seq run module and Z_BigDyeDirect DyeSet were selected.

## Analysis of methylation rates

Methylation rates were determined with the R-based tool ABSP, analysis of bisulfite sequencing PCR, implemented by Denoulet et al. [44]. This method relies on Bisulfite Sequencing PCR (BSP) originally developed by Frommer et al. [45]. BSP is comprised of the above-described steps: DNA denaturation, bisulfite conversion, PCR amplification, and sequencing. For this study, the direct-BSP approach was used, which means there was no cloning of PCR products. For the direct-BSP, signal ratios per CpG site were calculated by dividing the C signal by the sum of C signal and T signal [44]. It should be emphasized that the forward as well as the reverse files are used for analysis, which improves validity of the results [46]. The in-silico converted sense strand was used as the reference DNA sequence. Default thresholds remained the same, besides the following exceptions: (1) the maximum base-calling error probability was set 0.01 which resulted in a quality-value of 20, (2) the minimum ratio of primary peak to consider a position as non-mixed was set to 0.70, (3) the minimum percentage of non-mixed positions in the trimmed sequence to be considered non-mixed was changed to 70%,

**Table 2. Thermal cycler time and temperature for cycle sequencing.**

| Temperature | Time | |
|---|---|---|
| 37° | 15 min | |
| 80° | 2 min | |
| 96° | 1 min | |
| 96° | 10 sec | 25 cycles |
| 50° | 5 sec | |
| 60° | 4 min | |
| 4° | ∞ | |

(4) the minimum length of the trimmed sequence and (5) the minimum length of the aligned sequences were reduced to 20 bp, (6) threshold identity was set at 70, and (8) threshold conversion rate was changed to 0.80. Changes were made in accordance with standard values in SeqScape™ (Thermo Fisher Scientific, Waltham, Massachusetts).

A mean methylation rate of CpG sites −934, −924, and −901 was calculated, termed Methyl_3.

## Statistical analysis

Due to the heterogenous sample with respect to gender ($n = 55$ females) and age, a test for possible covariates was conducted prior to hypotheses testing. Furthermore, the relation between age and gender was evaluated because the mean age of males was two years higher ($M = 23.00$, $SD = 7.46$) compared to the mean age of females ($M = 20.65$, $SD = 2.04$). To test possible effects of gender, t-tests for independent samples were conducted with OXTR MT2 mean methylation rates, the neuroticism score, maternal and paternal care as well as the anxiety and the avoidance scale. Pearson correlations were preformed to test for the effect of age. No significant results were found, besides a significant negative correlation between age and paternal care, $r = -.242$, $p = .042$.

Four separate moderated meditation models were calculated, one for each hypothesis [a-d]. The following conditional effects were proposed:

(1) Neuroticism moderates the effect of perceived parental care, separately for father and mother, on anxious/avoidant attachment.

(2) Mean MT2 methylation levels mediate the effect of parental care on anxious/avoidant attachment.

(3) Neuroticism again moderates the effect of parental care on MT2 methylation levels.

The PROCESS macro for SPSS (v. 4.2) model 8 was used to test the conditional process model in Fig 2 [47]. Bootstrapping was set to 5000 samples and confidence intervals were 95%. HC3 (Davidson-MacKinnon) was set for heteroscedasticity-consistent inference and continuous variables that define products were centered.

All statistical analyses were performed with IBM SPSS Statistics for Windows version 29 (IBM Corp., Somers, NY, USA) with a significance level of $\alpha \le 0.05$. Bonferroni correction for hypotheses a and b as well as for hypotheses c and d was applied because of the high intercorrelation between attachment anxiety and attachment avoidance.

## Results

### Frequency distributions and intercorrelations of the relevant CpG sites

Methylation at the relevant CpG sites −901, −924, and −934 were generally higher compared to those reported in other studies. Fig 3 displays the frequency distributions of the relevant CpG sites. For illustration purposes, Fig 4 shows sequencing plots of four different and characteristic participants.

Based on the study by Danoff et al. [11] who postulated that the CpG sites −901, −924, and −934 are highly correlated, we checked for the intercorrelation between the three CpG sites (see Table 3). The intercorrelations obtained were not as high as stated by Danoff et al. [11]. Based on the literature, we still aggregated CpG sites −901, −924, and −934 into one variable (Methyl_3), because this cluster is most sensitive to experiences early in life and seems to be responsible for regulation of the OXTR gene expression [11]. So, Methyl_3 was used for further analyses. Additionally, the proposed moderated mediation model was already complex enough when focusing on Methyl_3; therefore, we did not want to calculate three different models, one for each CpG site.

### Descriptive statistics of relevant variables

Table 4 displays descriptive statistics as well as bivariate correlations between all variables in the present study.

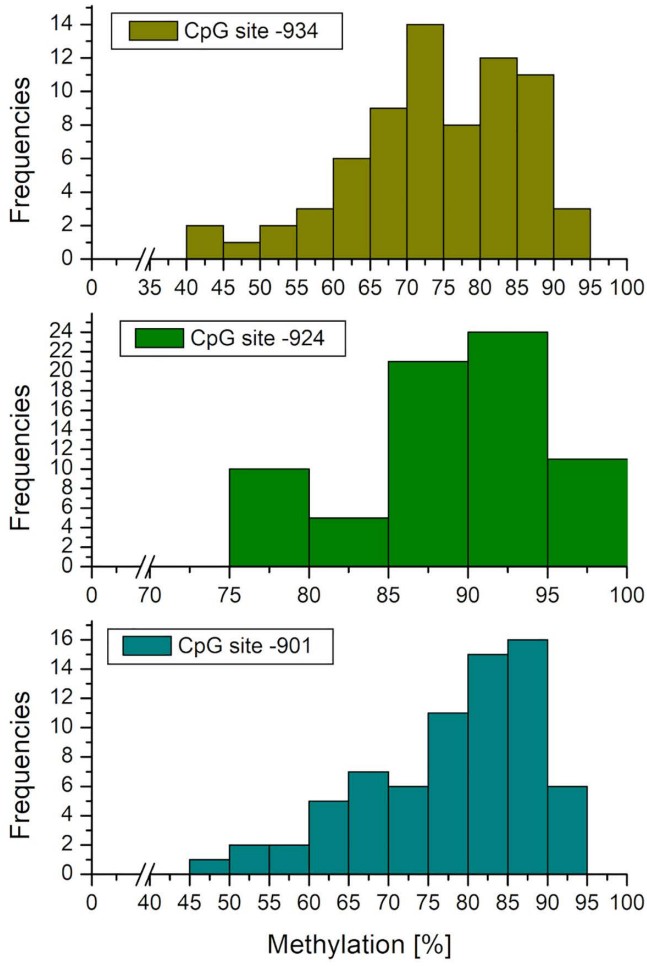

**Fig 3. Frequency distribution of the methylation [%] of the relevant CpG sites.** Numbering relative to transcription start site in the oxytocin receptor gene.

### Maternal care and attachment avoidance; mediated by MT2 methylation levels and moderated by neuroticism (hypothesis a)

Contrary to our hypothesis, neuroticism did not influence the negative relation between perceived maternal care and attachment avoidance, $F_{1,66}$(HC3) = 0.10, $p$=.754, $\Delta R^2$=.001, even though, the total model showed significance, $F_{4,66}$(HC3) = 9.94, $p$<.001, $\Delta R^2$=.383. Maternal care did not predict OXTR MT2 mean methylation levels and the mediation to attachment avoidance was not significant either, $F_{1,65}$(HC3) = 0.98, $p$=.327. However, neuroticism moderated the effect between perceived maternal care and OXTR MT2 methylation, $F$(1,67) = 5.33, $p$=.024, $\Delta R^2$=.095, 95% CI [−1.357, −0.298]. Neither perceived maternal care nor neuroticism levels had a direct effect on OXTR MT2 methylation (see Table 5). The conditional effects of the focal predictor showed that this effect was only significant among people with low neuroticism scores, $t$=−3.12, $p$=.003 (see Table 6 and Fig 5).

### Maternal care and attachment anxiety; mediated by MT2 methylation levels and moderated by neuroticism (hypothesis b)

No significant effects, neither direct nor indirect, on adult attachment anxiety were detected, even though the total model showed significance, $F_{4,66}$(HC3) = 12.82, $p$<.001, $\Delta R^2$=.452. MT2 methylation did not mediate the association between

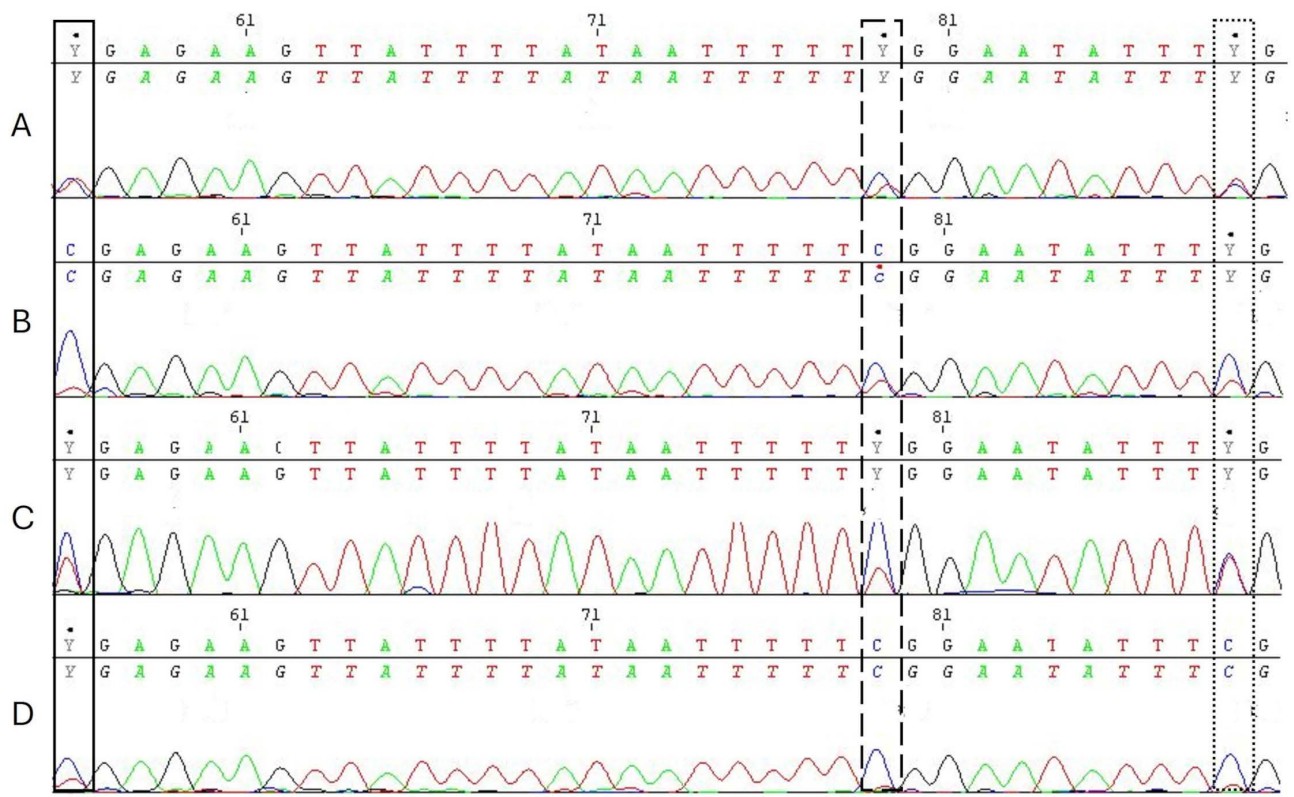

**Fig 4. Sequencing plots of the CpG sites −901, −924, and −934 (numbering relative to transcription start site).** Exemplary for four participants (A-D). Cytosine, C, indicates methylated CpG sites, while Y (cytosine or thymine) indicates partial methylation. The peak for mixed base detection was set to ≥ 25% in relation to the higher peak. CpG site -901 is displayed as solid line, CpG site -924 is displayed as dashed line, and CpG site -934 is shown by the dotted line.

**Table 3. Intercorrelations between CpG sites −901, −924, and −934, and their correlations with Methyl_3.**

|   | Variable | 1 | 2 | 3 | 4 |
|---|----------|---|---|---|---|
| 1 | CpG site −901 | – | .441*** | .366*** | .796*** |
| 2 | CpG site −924 |   | – | .465*** | .720*** |
| 3 | CpG site −934 |   |   | – | .816*** |
| 4 | Methyl_3 |   |   |   | – |

Note: ***$p < .001$

**Table 4. Descriptive statistics, means (M) and standard deviations (SD), and bivariate correlations between all relevant variables.**

|   | Variable | 1 | 2 | 3 | 4 | 5 | 6 | M (SD) |
|---|----------|---|---|---|---|---|---|--------|
| 1 | Maternal Care | – | .524*** | −.134 | −.117 | −.407*** | −.277** | 29.10 (5.83) |
| 2 | Paternal Care |   | – | −.109 | −.279* | −.292* | −.315** | 22.96 (8.54) |
| 3 | Methyl_3 |   |   | – | .080 | .240* | .106 | 80.24 (7.33) |
| 4 | Attachment Anxiety |   |   |   | – | .556*** | .659*** | 3.60 (0.65) |
| 5 | Attachment Avoidance |   |   |   |   | – | .534*** | 2.78 (0.59) |
| 6 | Neuroticism Score |   |   |   |   |   | – | 1.95 (0.57) |

Note: *$p < .05$ **$p < .01$ ***$p < .001$

**Table 5. Results of the moderated mediation in hypothesis a.**

| Predictor | Attachment Avoidance | | | | OXTR MT2 Mean Methylation Rates | | | |
|---|---|---|---|---|---|---|---|---|
| | b | SE (HC3) | LLCI | ULCI | b | SE (HC3) | LLCI | ULCI |
| Constant | 1.821** | 0.666 | 0.491 | 3.151 | 81.050*** | 0.872 | 79.309 | 82.790 |
| Maternal Care | −0.028** | 0.010 | −0.048 | −0.008 | −0.320 | 0.185 | −0.688 | 0.049 |
| Neuroticism Scores | 0.454*** | 0.105 | 0.244 | 0.663 | −0.515 | 1.553 | −3.616 | 2.585 |
| OXTR MT2 Methylation | 0.012 | 0.008 | −0.004 | 0.028 | | | | |
| Maternal Care x Neuroticism | 0.007 | 0.022 | −0.037 | 0.050 | 0.896* | 0.388 | 0.121 | 1.670 |
| R² | .383*** | | | | .118* | | | |

*Note:* Standardized regression coefficients are reported. Listwise $N = 71$, SE (HC3) = Davidson-MacKinnon standard error; LLCI = lower-level confidence interval; ULCI = upper-level confidence interval, Bootstrap sample size = 5000; confidence interval 95%; *$p < .05$ **$p < .01$ ***$p < .001$

**Table 6. Conditional effects of the focal predictor (maternal care) at different levels of neuroticism.**

| Conditional indirect effects at different levels of Neuroticism (M±1 SD) | b | SE (HC3) | LLCI | ULCI |
|---|---|---|---|---|
| −1 SD | −0.828** | 0.265 | −1.357 | −0.298 |
| M | −0.320 | 0.185 | −0.688 | 0.049 |
| +1 SD | 0.188 | 0.308 | −0.426 | 0.802 |

*Note:* SE (HC3) = Davidson-MacKinnon standard error; LLCI = lower-level confidence interval; ULCI = upper-level confidence interval; **$p < .01$

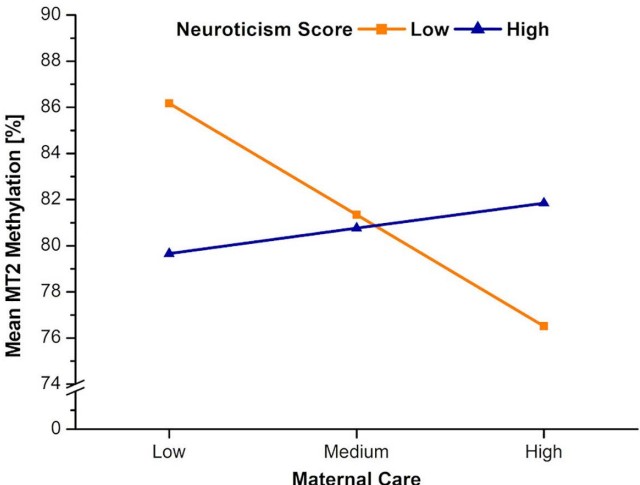

**Fig 5. Moderating effect of neuroticism on the effect of maternal care on MT2 mean methylation.** Mean methylation displayed in %. Only participants with low neuroticism showed significant effects.

maternal care and anxious attachment, $F_{1,65}$(HC3) = 0.48, $p = .490$, and neuroticism did not moderate the association, $F_{1,66}$(HC3) = 1.74, $p = .192$, $\Delta R^2 = .013$.

Solely the moderation effect of neuroticism level on the effect of perceived maternal care on OXTR MT2 methylation levels was significant, as shown in hypothesis a (see Table 5, right columns, Table 6, and Fig 5). For detailed statistics, see supporting information S1 Table.

## Paternal care and attachment avoidance and anxiety; mediated by MT2 methylation levels and moderated by neuroticism (hypotheses c and d)

For paternal care and avoidant attachment, the total model was significant, $F_{4,66}$(HC3) = 7.84, $p < .001$, $\Delta R^2 = .332$. However, neuroticism did not moderate the association between paternal care and attachment anxiety, $F_{1,66}$(HC3) = 0.02, $p = .888$, $\Delta R^2 = .000$, nor did MT2 methylation mediate the association, $F_{1,65}$(HC3) = 0.02, $p = .901$. Additionally, neuroticism did not moderate the association between paternal care and MT2 methylation, $F_{1,67}$(HC3) = 0.23, $p = .635$, $\Delta R^2 = .003$.

For paternal care and anxious attachment, the total model was significant as well, $F_{4,66}$(HC3) = 12.74, $p < .001$, $\Delta R^2 = .440$. However, neither did neuroticism moderate the association between paternal care and attachment anxiety, $F_{1,66}$(HC3) = 0.06, $p = .800$, $\Delta R^2 = .001$, nor did MT2 methylation mediate this association, $F_{1,65}$(HC3) = 0.21, $p = .646$. Additionally, as proposed in hypothesis c, neuroticism did not moderate the association between paternal care and MT2 methylation, $F_{1,67}$(HC3) = 0.23, $p = .635$, $\Delta R^2 = .003$.

For detailed statistics see supporting information S2 Table and S3 Table.

## Discussion

We investigated the associations among perceived parental care, OXTR MT2 methylation rates, adult anxious and avoidant attachment, and neuroticism. In contrast to previous studies, we looked at perceived maternal and paternal care separately, to obtain a more nuanced picture. Results point into the direction that only perceived maternal care was associated with adult attachment, more precisely, only with avoidant attachment. Neither mean OXTR MT2 methylation rate as a mediator, nor neuroticism as a moderator influenced the effect of perceived maternal care on adult avoidant attachment. However, neuroticism significantly altered the association between perceived maternal care and mean MT2 methylation rates: individuals with low levels of neuroticism showed increased levels of MT2 methylation levels, but only when perceived maternal care was low. When perceived maternal care was high, MT2 methylation rates were lower compared to when perceived maternal care was low. The next sections will discuss the main findings in detail. One has to keep in mind the exploratory and preliminary nature of the study.

## Perceived parental behavior and adult attachment

The results are in line with existing literature relating perceived parenting to attachment styles [23, 48]. Across all analyses, only maternal care was negatively associated with avoidance. This corresponds with a study by Fossati et al. [49]. They found a negative association between parental care and avoidant attachment but not anxious attachment. One explanation might be that individuals, who perceived their parents as less warm and understanding, develop less trusting attachment patterns and tend to avoid intimacy [49, 50]. As Fossati et al. [49] proposed, we looked at the underlying scales that compose attachment avoidance, namely Relationships as Secondary and Discomfort with Closeness, to get a more nuanced understanding of the results. Discomfort with Closeness closely resembles Hazan and Shaver's [51] concept of attachment avoidance, while Relationships as Secondary is more closely related to dismissing attachment as conceptualized by Bartholomew and Horowitz [25]. Conducting the same analysis as for hypothesis a both, Discomfort with Closeness, $F_{4,66}$(HC3) = 7.80, $p < .001$, $\Delta R^2 = .443$, and Relationships as Secondary, $F_{4,66}$(HC3) = 5.10, $p = .001$, $\Delta R^2 = .154$, revealed overall significant models. For Discomfort with Closeness no moderating effect of neuroticism on the association between parental care and MT2 methylation or the mediation of this association was shown. However, all predictors showed significant associations with Discomfort with Closeness (see supporting information S3 Table). For Relationships as Secondary, again, no moderation effect of neuroticism and no mediation effect of MT2 methylation was found. Only maternal care was a predictor for Relationships as Secondary (see supporting information S4 Table). Since Discomfort with Closeness resembles avoidant attachment, as conceptualized by Hazan and Shaver [51], the results are comparable to Ebner et al. [17]. We found a negative effect of MT2 methylation and Discomfort with Closeness in the

same direction as Ebner et al. [17]. Ein-Dor et al. [18] found a positive association between attachment avoidance and OXTR promotor methylation. Overall, this emphasizes the use of an even more nuanced approach when it comes to studies regarding attachment, because small differences in meaning, for example different questionnaires, can have an impact on the overall interpretation. Furthermore, the results illustrate the importance of methodological considerations to ensure comparability between studies. Additionally, one has to keep in mind that attachment styles are not categorial, but rather dimensional [38, 50]. Avoidance and anxiety show a weak correlation; therefore, people can score high (or low) on both, resulting in multiple different forms of insecure attachment, which makes it even more complex.

The result that perceived maternal care has a larger impact than paternal care, was not surprising; however, we did not expect to find a total lack of effects for paternal care in our sample. The different influence mothers have in relation to fathers is well established, for example, in the model of transgenerational transmission, as stated by Verhage et al. [22]. This can be due to the fact that, traditionally, mothers were the main caregivers, while fathers went to work to provide for the family. In recent years, this family model was outpaced by more diverse models, for example stay-at-home fathers or rainbow families. Therefore, in future studies participants should be asked to characterize their family model, to make sure the results can be interpreted correctly. Yaffe [52] revealed in his systematic review that a majority of studies found significant differences between mothers and fathers in overall parenting. Namely, mothers tend to be more caring and warm, but also controlling, whereas fathers are more harsh and restrictive. This is in line with our results: maternal care was rated higher than paternal care. We did not consider the control scale of the PBI. Additionally, several studies found a gender of the parent x gender of the offspring interaction. For example, Huang et al. [53] revealed that male descendants tend to perceives mother and father as more authoritarian (emotionally distant and strictly controlling – less favorable), while daughters tend to rate their parents as more authoritative (highly demanding and responsive – overall favorable). Because of the unbalanced gender distribution of the sample, we did not conduct gender of the parent x gender of the offspring interactions. However, future studies should consider gender-role theories. Overall, the null finding for paternal care is still somewhat unexpected and has to be examined in greater detail.

## Methylation and (childhood) experiences

The effect of negative childhood experiences on MT2 methylation has been shown in numerous studies (e.g., [15, 54]). So far, there are only two studies looking at the effect of positive experiences in childhood, more precisely high perceived maternal care, and its effects on OXTR methylation. Unternaehrer et al. [55] revealed a direct effect of maternal care, measured by the PBI, on OXTR methylation. The results of our study do not match the results by Unternaehrer et al. [55], since no direct effect of maternal care on OXTR methylation was found, which might be due to the fact that the region of interests slightly differed. Our study focused on the MT2 region, a 406 bp long region at the beginning of the CpG island in the OXTR, whereas Unternaehrer et al. [55] examined the effects of parental care on regions covering mainly exon 3, which is located at the end of the CpG island. Consequently, future research should be precise when presenting results with regard to oxytocin, because differences in the region of interest can have an impact on the results and their interpretation. The other study by Krol, Moulder et al. [14] also found a direct effect of maternal care on the infant's OXTR methylation. They conducted a longitudinal study, following infants and their mothers from the age of five months up to 18 months. Mothers completed questionnaires and mother-infant dyads were observed during play behavior at five months of age. Within this sensitive period, the way the mother behaved towards its infant affected OXTR methylation in the infant – the infant's methylation levels changed in relation to maternal care. Infants who experienced more maternal care at five months had reduced methylation levels with 18 months, which speaks in favor of a sensitive period in development around the age of five months. Our study was conducted on psychology students with a mean age of about 21 years of age. The PBI aims to retrospectively and subjectively assess parental behavior separately for mother and father up to 16 years of age [26]. Although there is literature [56], showing that the PBI displays good stability over the course of 20 years, the participants subjective perception of their parents might have changed in late adolescence and early adulthood.

Furthermore, there might have been major life-events in this period, which could have affected, even subconsciously, the evaluation. Another possible explanation might be that a reference person changes over the lifespan. During childhood, parents are the main influence for children, but during adolescence, the focus shifts from parents to peers as role models [57]. Since methylation is a dynamic process, it might be the case that experiences besides parental engagement have an effect on the level of methylation, which confounded the initial association found in a study conducted with children [14, 55]. In our study, the association between maternal care and MT2 methylation was negative, as expected, but not significant, $p = .088$; however, the trend went into the assumed direction. Additionally, it should be mentioned that MT2 methylation shared a positive association with attachment avoidance, even though this path was not significant (see Table 5). This trend is in accordance with the literature as well, because increased methylation may lead to decreased OXTR function, with is related to insecure attachment [17, 18].

Next, the potentially different influences of experiences within and outside of the family will be discussed. To differentiate between experiences within a family and experiences outside of the family, Li et al. [58] conducted a study on monozygotic and dizygotic twins. They revealed that, at birth, monozygotic and dizygotic twins had a methylation rate correlation of zero, comparable to unrelated samples. However, the longer twin pairs lived together, the more similar their methylation rates became, with a peak at the age of 18 years. After leaving the shared environment, the correlation of methylation rates decreased dramatically, which emphasizes the effects of early childhood but also the fact that non-shared environment in adulthood does have an influence on methylation [58]. This is in line with the assumption that experiences the participants gain in adolescents or early adulthood might, to some extent, override, or at least strongly influence, the initial experiences in childhood. As a result, methylation rates might have changed. On the other side, Dunn et al. [59] emphasized that the timing of exposure to a certain event plays a significant role for DNA methylation. Their results showed that in early childhood (before the age of 3 years) all types of adverse experiences had an effect on DNA methylation, whereas at an older age only severe adverse experiences influence methylation. This sensitive period in early years seems to be maximal susceptible to environmental influences and, moreover, gene specific [59]. No study so far has looked at positive experiences; however, it can be assumed that children in early developmental stages are more sensitive to negative as well as positive experiences, characterizing this peak in plasticity. Thus, the PBI might not be the appropriate survey method, since active memory recall starts around the age of two and a half years [60]. At this point the developmental phase in which all severity levels of experiences have an influence is, according to Dunn et al. [59], almost over.

Concluding, our results show first indications for an interaction of maternal care, MT2 methylation, and attachment which goes in the expected direction but is not significant. A possible explanation might be the large time gap between the predictor, maternal care, and the measuring of this construct. Furthermore, life-events between childhood and the time of the survey should have been accounted. Nevertheless, it is probable that experiences in childhood have an influence on MT2 methylation rates.

### The role of neuroticism

The notion that adult attachment styles are influenced by experiences in childhood goes back to Bowlby [61]. However, to the best of our knowledge, no study so far accounted for the child's personality, which has a genetic contribution of about 40% [29]. Therefore, it should be considered, because the way parents act towards their child also depends on how the child acts towards them (active gene x environment correlation). Besides genetics, early environmental influences, especially through parental behavior, might impact personality. Perceived parental behavior and personality may interactively contribute to changes in OXTR methylation.

For all analyses, neuroticism was positively correlated with attachment avoidance and attachment anxiety. This is in line with results by Hannuschke et al. [62], showing that high neuroticism scores were associated with decreased subjective relationship satisfaction, and results by Noftle and Shaver [63], who found a positive association between neuroticism and insecure attachment. According to the Differential Susceptibility Model [64], people respond differently to environmental

influences. Individuals who are sensitive and responsive flourish when they experience positive environmental influences, but they also have more negative outcomes when experiencing negative impacts. People being less sensitive, vary less in response to different environmental influences [65]. A similar concept, the sensory processing sensitivity (SPS) overlaps to some extent with the Differential Susceptibility Model; however, the SPS states that differences in environmental susceptibility are due to inter-individual differences in temperament and personality, characterized by a greater depth of information processing and a highly sensitive nervous system, sharing a positive correlation with neuroticism [66]. People high in SPS are easily overstimulated, whereas people with low SPS scores actively search for social stimulation [67]. With respect to the present study, participants with low neuroticism scores might have actively searched for their parents' influential behavior to compensate for their lower sensitivity, making them react more to both high and low maternal care (Fig 5).

One point to consider is that we did not pre-screen our sample for any exclusion criteria, for example unreported psychiatric conditions. The main reason for that is our focus on neuroticism as a key variable in this study. Many psychiatric conditions, for example anxiety or depression, are linked to neuroticism [68]. Our goal was to have as much variance in neuroticism as possible. Additionally, excluding participants with high neuroticism scores would limit generalizability, since it does not represent the general population.

## Methylation and personality

Wilson and Durbin [69] showed that children with low neuroticism scores experienced more warm and structured parenting and overall more positive interactions, while children with high neuroticism scores had less responsive parents. In our exploratory study, the combination of maternal care and low neuroticism went in the expected direction, namely people with low neuroticism and high maternal care displayed decreased methylation, which is assumed to be favorable, whereas low neuroticism and low maternal care was associated with high MT2 methylation (see Fig 5). The opposite trend was shown for people with high neuroticism scores. For medium maternal care, there was almost no difference in MT2 methylation rates between people with high versus people with low neuroticism scores. To sum up, the child's personality may act as the moderating factor between perceived maternal care and MT2 methylation.

However, not only the way the child behaves towards their mother but the other way around is also important. Because of the high genetic component of personality [29] and the fact that the whole sample was reared by their biological parents, it can be assumed that mothers, to some extent, share personality characteristics with their child. Smith et al. [70] conducted a study with 30–36 months old toddlers and their mothers, claiming that toddlers with more warm and responsive mothers, which is an indicator low neuroticism scores in the mothers, displayed more positive emotions, which in turn facilitate positive interactions between mother and child [71]. This emphasizes the importance of maternal parenting behavior. To conclude, we assume people with low neuroticism scores to have biological mothers with low neuroticism scores as well, which facilitates their interaction, resulting in a decrease of MT2 methylation, when mothers display favorable parenting behavior and increased MT2 methylation, when mothers are less caring. To support this assumption, future studies should collect more information about mothers (or the primary caregiver being studied), for example childhood experiences, life-events, or attachment style.

## Limitations & future directions

One strength of the present study is that we used both the forward and the reverse primer for methylation analysis, as recommended by Rubino et al. [46]. However, there are limitations regarding the analysis. First, we did not use triplets, as proposed by the ABSP tool, but we checked for reliability with a subset of samples. Except for one sample, all samples showed good re-test reliability (see supporting information S6 Table). Another strength of the study was the theory-based approach. Unlike genome-wide association studies or epigenome-wide association studies our research question was based on previous studies, for example the focus on 3'-sites of the MT2 region, whose functional significance was shown [11]. The assessment of

neuroticism, attachment, and perceived parental behavior relied solely on self-reports. We chose the ASQ to gather information about adult attachment. According to Jewell et al. [72] there are attachment measure which have more adequate measurement properties, for example the Inventory of Parent and Peer Attachment. Nevertheless, because of the longitudinal design of the data collection – the same questionnaires were applicated to first year psychology students every year over a time span for at least 7 years, we used the ASQ, even though it has limitations, namely not sufficient structural validity [72]. For personality, self-reports are the main base of information; however, there are studies suggesting the use of informant-reports from partner, close friends, or family to avoid biases [73]. For parental behavior, parents' evaluation of the atmosphere within the family might be interesting as well. Additionally, if siblings are present, their assessment might be of relevance to get a more comprehensive picture. Despite the PBI being a widely used and well-established questionnaire, our sample had a wide age (18–48 years). It has to be pointed out that the majority of participants were between 18 and 28 years, only one participant was 48 years of age. Wilhelm and colleagues [56] conducted a study, showing that the perception of parental care is stable over a 20 year period. Therefore, the assessment of parental care of the 48-year-old participant might not be as accurate as the assessment of younger participant. However, due to the already small sample size, we decided not to exclude this participant.

In future studies, data regarding the actual family model should be collected, for example, who was the primary care-giver, how many siblings are in a family, whether it is a patchwork or rainbow family, and so forth. Other potential influences, like certain life-events, birth order, or socioeconomic status, should be queried and put into relation. A much bigger sample is necessary to divide people into different groups depending on their family model. It would be interesting to see if the primary caregiver has more impact or if the impact depends on the gender of the child or the gender of the caregiver. Several studies found an interaction between MT2 methylation and OXTR rs53576 genotype (for a review, see Prata and Silva [74]). Our study did not find any interactional effects, which might be due to the small sample size. A bigger sample also provides the chance to include primary caregivers divided into certain attachment styles, to further elaborate on the theory of intergenerational transmission. In order to do that the primary caregiver's childhood experiences, methylation status and attachment styles should be collected. Furthermore, the time frame of the PBI is presumably too broad to get a nuanced picture of the effects of parental care on methylation. Future studies should narrow the time frame and, more importantly, should consider sensitive periods in development [59]. Finally, methylation analyses are still new in the field of psychology and were introduced with high euphoria. However, many important questions are still unanswered. First, most approaches are correlational in nature, meaning that psychological constructs (e.g., parental behavior) cannot be interpreted as causing methylation. Second, methylation patterns might differ between certain cell types. Although a correlation between methylation patterns in buccal cells and neurons has been described [41], functional differences can be expected. The dynamics of methylation and demethylation are completely unknown but of course extremely important when looking at critical experiences and their ongoing impact. Longitudinal studies are needed to get the required information for changes in methylation or, more general, for stability of methylation patterns.

The large drop-out, the initial sample had $N = 367$ participants, whereas the final sample consisted of $N = 71$, is another limitation of the present study, because systematic selection bias could have happened. Most of the drop-out was due to incomplete questionnaires ($n = 197$). Participants with poor relationships with their parents might have terminated the survey, because they simply did not want to share information about their relationship with their parents. Therefore, the final sample might be biased on people who like to share information about their relationship with their parents, which limits generalizability. Another limitation regarding the sample is its homogeneity. The sample mainly consists of Caucasian females about 21 years of age, who study psychology. This is a really narrow population group; therefore, results have to be interpreted with caution. However, Ferber [75], states that a convenience sample is sufficient for preliminary and exploratory studies to get a first overview before spending a lot of money on a representative sample. Finally, the small sample size significantly impacts the statistical power of the moderated mediation model. This has several implications. First, due to the insufficient data small differences could not be detected. Second, the significant association, which was found, might not reflect a true effect. Third, the reproducibility of results might be limited, especially when conducting the

study with a more heterogeneous group. Therefore, the present results are only exploratory and have to be replicated in a much larger and more diverse sample.

## Conclusion & practical significance

This exploratory study provides first preliminary insights into the cumulative effect of positive early experiences, DNA methylation in the OXTR MT2 region, and neuroticism on adult attachment. The overall models incorporating anxious and avoidant attachment as well as the distinction between the effect of maternal and paternal care, were not significant. Yet, initial results show that neuroticism significantly moderates the effect of maternal care on OXTR MT2 methylation. This emphasizes the importance of gene x environment interactions for methylation studies: individual differences in children can influence the association between parental behavior and methylation levels. However, many more influences, like different family models or the stability of methylation patterns, should be addressed in future studies to get an as comprehensive picture as possible of methylation and attachment. Future studies are encouraged to change the narrative from "experiences shape certain outcomes", to a broader perspective. Experiences in childhood and adolescence do have an effect on behavior later in life, perhaps through epigenetic changes, but more importantly, one's personality shapes the way experiences are perceived and processed. Therefore, if an experience is associated with more or less favorable outcomes in the long run, it does not necessarily only rely on the experience itself, but rather on an interplay between personality, genetics, experiences, and many more variables. Future studies have the task of detecting as many of these variables as possible to further disentangle complex behaviors.

## Supporting information

**S1 Table. Results of the moderated mediation in hypothesis b.** Maternal care and attachment anxiety: mediated by MT2 methylation levels and moderated by neuroticism.
(DOCX)

**S2 Table. Results of the moderated mediation in hypothesis c.** Paternal care and attachment avoidance: mediated by MT2 methylation levels and moderated by neuroticism.
(DOCX)

**S3 Table. Results of the moderated mediation in hypothesis d.** Paternal care and attachment anxiety: mediated by MT2 methylation levels and moderated by neuroticism.
(DOCX)

**S4 Table. Results of the moderated mediation for maternal care and Discomfort with Closeness.** Mediated by MT2 methylation levels and moderated by neuroticism.
(DOCX)

**S5 Table. Results of the moderated mediation for maternal care and Relationships as Secondary.** Mediated by MT2 methylation levels and moderated by neuroticism.
(DOCX)

**S6 Table. Method check.** Bivariate correlation analysis for a subset of samples ($n = 9$) re-analyzed with a four-month gap in between the analyses.
(DOCX)

## Acknowledgments

We are greatly indebted to Nicole Tscherney for running the genetic analyses and in all our subjects in providing the data.

## Author contributions

**Conceptualization:** Laura Geißert.

**Formal analysis:** Laura Geißert.

**Investigation:** Laura Geißert.

**Methodology:** Laura Geißert, Juergen Hennig.

**Resources:** Juergen Hennig.

**Supervision:** Juergen Hennig.

**Visualization:** Laura Geißert.

**Writing – original draft:** Laura Geißert.

**Writing – review & editing:** Laura Geißert, Juergen Hennig.

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
