## [Decision Letter · Decision Letter 0]

6 Oct 2025

PONE-D-25-33081Parental behavior, adult attachment, and DNA methylation of the MT2 oxytocin receptor gene region – The moderating role of neuroticismPLOS ONE

Dear Dr. Geißert,

Thank you for submitting your manuscript to PLOS ONE. After careful consideration, we feel that it has merit but does not fully meet PLOS ONE’s publication criteria as it currently stands. Therefore, we invite you to submit a revised version of the manuscript that addresses the points raised during the review process.

This manuscript requires major revisions addressing methodological transparency, statistical limitations, sample representativeness, and interpretive caution. The exploratory nature of the study must be explicitly acknowledged throughout, with more measured claims regarding transgenerational transmission, clearer discussion of practical significance, and enhanced clinical rigor in future studies. Address sample recruitment rationale, missing intergenerational and contextual data, and reframe conclusions appropriately.

We look forward to receiving your revised manuscript.

Kind regards,

Rei Akaishi

Academic Editor

PLOS ONE

Journal Requirements:

Additional Editor Comments:

Both reviewers acknowledge the manuscript addresses an important and timely topic in neuroscience concerning oxytocin receptor gene methylation, maternal care, and attachment. Reviewer 1 considers the biological methodology strong, while Reviewer 2 finds the integration of genetic, psychological, and environmental perspectives innovative and relevant. However, both reviewers agree the study should be framed as exploratory and preliminary, with significant revisions necessary before publication.

INTRODUCTION

Structure and Length: Reviewer 2 notes the introduction is overly lengthy with redundant sections (e.g., repeated content on methylation and OXTR polymorphisms). Streamlining is needed for improved readability.

Rationale Clarity: Reviewer 2 indicates the justification for focusing on the MT2 region and selected CpG sites needs stronger articulation.

Framing of Claims: Reviewer 2 notes the proposed mediating role of methylation is ambitious given sample size limitations and should be presented more cautiously.

Transgenerational Transmission Statement: Both reviewers express concern about oversimplification of transgenerational attachment transmission. Reviewer 2 notes the statement is overly direct and oversimplifies a complex concept, requiring reformulation with authoritative references. Reviewer 1 similarly emphasizes this concept is "considerably more complex and nuanced" than presented.

Study Classification: Reviewer 2 recommends explicitly stating this is an exploratory, preliminary investigation throughout the manuscript.

METHODS

Sample Recruitment and Characterization:

Lack of Inclusion/Exclusion Criteria: Both reviewers identify this as a major concern. Reviewer 1 notes participants were recruited from university settings without specified criteria, potentially including individuals with unreported psychiatric conditions (anxiety, depression, autism spectrum disorders). Reviewer 2 emphasizes that studies typically exclude cognitive disabilities, intellectual impairments, or psychopathology, and requests clarification for the rationale behind this decision.

Sample Attrition and Bias: Reviewer 2 notes the final sample (N=71) represents significant reduction from original cohort (N=367), with potential for systematic selection bias that should be explicitly acknowledged.

Statistical Power: Reviewer 2 indicates sample size is likely insufficient for the complex moderated mediation models employed and requires discussion as a limitation.

Sample Homogeneity and Generalizability: Both reviewers emphasize significant generalizability concerns. Reviewer 2 notes the sample consists of psychology students, predominantly female, all Caucasian, mean age 21, substantially restricting generalizability. Reviewer 1 similarly identifies the university recruitment setting and notes possible overrepresentation of single-parent families as limitations.

Age Range Variability: Both reviewers note concerns about age range. Reviewer 1 identifies wide age range (18-48 years) as introducing potential confounding variability. Reviewer 2 emphasizes this creates recall bias concerns for the retrospective Parental Bonding Instrument.

Missing Data and Contextual Factors:

Intergenerational Data Gap: Reviewer 1 identifies lack of information about participants' mothers' childhood experiences, attachment histories, and maternal methylation patterns as crucial missing context for interpreting offspring outcomes.

Confounding Variables: Reviewer 2 notes relevant contextual factors (family structure, socioeconomic status, life events) were not collected and should be addressed as a limitation.

Measurement Issues:

Instrument Selection: Reviewer 2 notes attachment was assessed exclusively through self-report ASQ and requests justification for choosing this measure when more statistically robust instruments are available.

CpG Site Aggregation: Reviewer 2 indicates aggregation of three CpG sites (-901, -924, -934) into composite index (Methyl_3) is not fully justified, especially given weak intercorrelations, and requires clarification of rationale.

RESULTS

Statistical Correction: Reviewer 2 notes Bonferroni correction was applied only to selected hypotheses, with unclear rationale requiring explicit explanation.

DISCUSSION

Length and Focus: Reviewer 2 finds the discussion overly long and occasionally digressive (e.g., lines 454-461), requiring more concise, focused narrative.

Interpretation of Null Results: Reviewer 2 notes tendency to justify null findings with excessive speculation (small sample, recall bias, genomic region differences) and recommends more critical and balanced tone.

Causal Language: Reviewer 2 emphasizes the cross-sectional design does not support causal inference and multiple statements require revision to avoid causal language.

Paternal Care Findings: Reviewer 2 notes the role of paternal care is somewhat downplayed, with absence of significant effects needing direct acknowledgment rather than extensive speculation.

Conclusion Framing: Both reviewers emphasize the need for more cautious framing. Reviewer 2 states findings should emphasize exploratory evidence rather than definitive support for mediation/moderation model. Reviewer 1 notes the results are interesting but preliminary.

Results Repetition: Reviewer 2 recommends not repeating numerical results in discussion for improved readability.

Study Nature: Reviewer 2 states the exploratory and preliminary nature must be explicitly acknowledged throughout.

Practical Significance: Reviewer 2 requests clearer discussion of what contribution these data make to the field and implications for future research.

FUTURE DIRECTIONS

Reviewer 1 suggests:

Continue longitudinal follow-up with more rigorous clinical characterization

Investigate relationship between neuroticism and temperamental inhibition (RGS2 gene), particularly regarding social anxiety phenotypes

Recruit more targeted samples (e.g., offspring of mothers with documented childhood trauma and known oxytocin receptor methylation patterns) to test specific hypotheses about intergenerational transmission

CONCLUSION

Tone: Both reviewers recommend more measured and cautious tone to reflect the exploratory and preliminary nature of the study.

Reviewers' comments:

Reviewer's Responses to Questions

**Comments to the Author**

1. Is the manuscript technically sound, and do the data support the conclusions?

Reviewer #1: Yes

Reviewer #2: Partly

2. Has the statistical analysis been performed appropriately and rigorously? 

Reviewer #1: Yes

Reviewer #2: Yes

3. Have the authors made all data underlying the findings in their manuscript fully available?

Reviewer #1: Yes

Reviewer #2: Yes

4. Is the manuscript presented in an intelligible fashion and written in standard English?

Reviewer #1: Yes

Reviewer #2: Yes

5. Review Comments to the Author

Reviewer #1: The data support the conclusions; the statistical analysis were conducted rigorously; the data are available and the manuscript is written in standard English. I have requested that major revisions be made to allow the paper to be published.

Reviewer #2: First, I can confirm this article is well-written, interesting, well explained and explore such an important topic in Science (Oxytocin receptor gene with methylations, maternal care and attachment style that I partially studied in Neurosciences in Claude Bernard Lyon University in 2018, Neurosciences Research Master, article send by email).

Here are my observations about limitations in that study and I hope some new future directions too.

In my opinion, these following points should be written in a paragraph to highlight the importance of that article.

1. The studied population is recruited in University with no inclusive or exclusive criteria. It is about general population (probably with many monoparenting families) that is a limited point. Indeed, we can find some people presenting with any disorder (student with autism rarely or anxiety / depression frequently that is an evidence nowadays). In Universities, we know that some young people can suffer from anxiety and emotional disorder. Moreover there is a high variability in the sample age here (18 - 48) to take into account.

2. Another clinical shortcomings is we don t know the childood mother and their infancy....and what s about their own methylations that may help us to understand this study.

3. Even if we can see shortcomings / gaps / lacks in the clinical methodology (biological methodology seems to be perfect+++), the results are interesting and logical to my mind. The initial hypotheses (p.14, p.15, p16.) should be confirmed in a particular sample, for example people who had mothers presenting with childhood trauma and higher methylations Oxytocin receptors. It is my hypothesis. Moreover, neuroticism is also often linked to inhibition temperament (RGS2 gene) as a well known characteristic which lead to social phobia for example (see my medical thesis online, send by email). It would be very interesting to explore that key point in the future.

The need is to continue to make this longitudinal study, reinforcing clinical rigourous scientific methodology in my opinion.

Remark : We found same results in a general population sample.

In our article published in Lausanne University (Tible O, von Gunten A.), we found in different sample as follows :

rs53576 polymorphism was found in MCI, rather than in the general population. The rs53576 was linked with separation anxiety in adulthood,post-traumatic stress disorder and depression, mediated by stress reactivityand neuroticism [15-17,22]. Our results are in the line with these observations,and in favour of our hypothesis that rs53576 OTrx may be a risk factor to affective disorder. The rs53576 genotype “may present a greater biological sensitivity as well as stress reactivity” according to Chang et al. [22]. This polymorphism would prevent the RNA polymerase from recognizing the RNA chain. Accordingly, a single SNP can predict most of the variance inOT receptor expression in specific brain regions [24]. Indeed, rs 53576 (AG+or AA+ carriers) is less expressed in the cerebral regions which are involvedin social behaviour related to OT. Thus, it leads to decrease OT activity inspecific brain structures (accumbens nuclei and limbic structures involved inJ Behav Neurosci Vol 2 No 1 November 2018social attachment), leading to anxiety through the high amygdala’s reactivity[6,23,25]. Although we have found no link between rs53576 (AG+ carrier)and personality traits (neuroticism, extraversion), like other authors whodidn’t find any correlation between the rs53576 (AG+) and the Big Fiveinventory [25,26]

From what precedes, you surely have really future directions to find out new important results in that cohort in the future.

Dr Olivier Tible

See

1.

Psychoneuroendocrinology.

Author manuscript; available in PMC: 2015 May 1.

Published in final edited form as: Psychoneuroendocrinology. 2014 Jan 30;43:11–19. doi: 10.1016/j.psyneuen.2014.01.012

Oxytocin Receptor Gene Polymorphism (rs53576) Moderates the Intergenerational Transmission of Depression

2.https://iris.unil.ch/handle/iris/94854

URL éditeur

https://www.pulsus.com/abstract/oxytocin-receptor-polymorphisms-and-attachment-style-in-patients-with-cognitive-impairment-and-affective-symptoms-4970.html

6. PLOS authors have the option to publish the peer review history of their article (what does this mean? ). If published, this will include your full peer review and any attached files.

**Do you want your identity to be public for this peer review?** For information about this choice, including consent withdrawal, please see our Privacy Policy .

Reviewer #1: No

Reviewer #2: **Yes:** Olivier Tible-Siri M.D.

---

## [Author Response · Author response to Decision Letter 1]

8 Dec 2025

Dear Reviewer 1, Dear Olivier Tible-Siri,

thank you for the opportunity to revise my manuscript. I considered all comments you have made and I am grateful to get another chance to publish my work. A detailled overview of how I incorporated your remarks is in the file Response to Reviewer.

Kind Regards,

Laura Geißert

---

## [Decision Letter · Decision Letter 1]

2 Jan 2026

Parental behavior, adult attachment, and DNA methylation of the MT2 oxytocin receptor gene region – The moderating role of neuroticism

PONE-D-25-33081R1

Dear Dr. Geißert,

We’re pleased to inform you that your manuscript has been judged scientifically suitable for publication and will be formally accepted for publication once it meets all outstanding technical requirements.

Kind regards,

Rei Akaishi

Academic Editor

PLOS One

Additional Editor Comments (optional):

The authors addressed the major issues raised by reviewers.

Reviewers' comments:

Reviewer's Responses to Questions

**Comments to the Author**

1. If the authors have adequately addressed your comments raised in a previous round of review and you feel that this manuscript is now acceptable for publication, you may indicate that here to bypass the “Comments to the Author” section, enter your conflict of interest statement in the “Confidential to Editor” section, and submit your "Accept" recommendation.

Reviewer #1: All comments have been addressed

Reviewer #2: All comments have been addressed

2. Is the manuscript technically sound, and do the data support the conclusions?

Reviewer #1: Yes

Reviewer #2: Partly

3. Has the statistical analysis been performed appropriately and rigorously? 

Reviewer #1: Yes

Reviewer #2: Yes

4. Have the authors made all data underlying the findings in their manuscript fully available?

Reviewer #1: Yes

Reviewer #2: Yes

5. Is the manuscript presented in an intelligible fashion and written in standard English?

Reviewer #1: Yes

Reviewer #2: Yes

6. Review Comments to the Author

Reviewer #1: The authors have responded appropriately to the comments made and the version of the manuscript is now acceptable for publication

Reviewer #2: Dear Colleagues

You should write a last paragraph. Indeed we can conclude here that it is so important to study oxytoxin receptors and attachment styles in psychiatric Pathologies and in Cancer and Depression for example to find out links between infancy traumas and methylations OX receptors in one hand, and otherwise links between attachment style / temperamental characteristics and emotional disorders within some Life Style patients.

You can see our pilot study which confirm some hypotheses in older age.

See

Oxytocin receptor polymorphisms and attachment style in patients with cognitives impairment and affective symptoms.

J Behav Neurosci Vol 2 No 1 November 2018

Best regards

Olivier Tible

7. PLOS authors have the option to publish the peer review history of their article (what does this mean? ). If published, this will include your full peer review and any attached files.

**Do you want your identity to be public for this peer review?** For information about this choice, including consent withdrawal, please see our Privacy Policy .

Reviewer #1: **Yes:** Chiara Pesca

Reviewer #2: **Yes:** Olivier Tible, M.D.,

Psychiatrist Medical Doctor, Medical Chief in Psychosomatics Clinics.

Oxytocin receptor polymorphisms and attachment style in patients with cognitives impairment and affective symptoms.

J Behav Neurosci Vol 2 No 1 November 2018

---

## [Editor Report · Acceptance letter]

PONE-D-25-33081R1

PLOS One

Dear Dr. Geißert,

I'm pleased to inform you that your manuscript has been deemed suitable for publication in PLOS One. Congratulations! Your manuscript is now being handed over to our production team.

Kind regards,

on behalf of

Dr. Rei Akaishi

Academic Editor

PLOS One